# Transition of Carbon Nanotube Sheets from Hydrophobicity to Hydrophilicity by Facile Electrochemical Wetting

**DOI:** 10.3390/nano13212834

**Published:** 2023-10-26

**Authors:** Myoungeun Oh, Hyunji Seo, Jimin Choi, Jun Ho Noh, Juwan Kim, Joonhyeon Jeon, Changsoon Choi

**Affiliations:** 1Department of Energy and Materials Engineering, Dongguk University, 30 Pildong-ro, 1-gil, Jung-gu, Seoul 04620, Republic of Korea; abcdeun@dongguk.edu (M.O.); amy0527330@gmail.com (H.S.); james010828@gmail.com (J.C.); shwnsgh15@dongguk.edu (J.H.N.); kjw3328@dongguk.edu (J.K.); 2Department of Advanced Battery Convergence Engineering, Dongguk University, 30 Pildong-ro, 1-gil, Jung-gu, Seoul 04620, Republic of Korea; 3Division of Electronics & Electronical Engineering, Dongguk University–Seoul, 30 Pildong-ro 1-gil, Jung-gu, Seoul 04620, Republic of Korea

**Keywords:** electrochemical wetting, carbon nanotube (CNT), supercapacitor, flexibility, wearable

## Abstract

The present study delves into the transformative effects of electrochemical oxidation on the hydrophobic-to-hydrophilic transition of carbon nanotube (CNT) sheets. The paper elucidates the inherent advantages of CNT sheets, such as high electrical conductivity and mechanical strength, and contrasts them with the limitations posed by their hydrophobic nature. A comprehensive investigation is conducted to demonstrate the efficacy of electrochemical oxidation treatment in modifying the surface properties of CNT sheets, thereby making them hydrophilic. The study reveals that the treatment not only is cost-effective and time-efficient compared to traditional plasma treatment methods but also results in a significant decrease in water contact angle. Mechanistic insights into the hydrophilic transition are provided, emphasizing the role of oxygen-containing functional groups introduced during the electrochemical oxidation process.

## 1. Introduction

Forest-spun carbon nanotube (CNT) sheets exhibit multifunctionality, making them the material of choice for a wide array of flexible electronics [1]. Specifically, their low sheet resistivity (approximately 700 Ω per square in the drawing direction) and high transmittance (ranging from 95% to 98.5% for various radiation types) are highly applicable to displays, video recorders, solar cells, and solid-state lighting [1]. Moreover, the microstructure of the CNT sheet is like a network, consisting of primary CNT bundles aligned in the drawing direction and secondary branches interconnecting these main bundles. This unique network structure is formed by cooperatively rotating the CNTs in vertically oriented nanotube arrays. The networked architecture allows high retention of electrical conductivity when the sheets are mechanically bent or deformed [1], a property not found in conventional transparent conductors like indium tin oxide (ITO).

Despite such advantages, one critical limitation of CNT sheets is their intrinsic hydrophobicity originating from the graphitic carbon–carbon bonding [2,3,4]. Therefore, hydrophilic modification of the CNT sheets is often a critical requirement for wider applications demanding enhanced wettability and surface interactions with aqueous solutions. For instance, hydrophilic CNT sheets facilitate efficient adsorption of contaminants, thereby improving the overall performances of filtration systems. In biomedical applications, hydrophilic CNT sheets are preferred for their compatibility with biological fluids, which is essential for drug delivery or tissue engineering. Hydrophilic surfaces also promote better dispersion in polymer matrices, enhancing the mechanical and thermal properties of CNT-based composites. Most importantly, in electrochemical applications such as supercapacitors and batteries, hydrophilic CNT sheets can improve electrolyte wettability, leading to enhanced ion transport and improved charge storage capability thereof. Therefore, facile transition of CNT sheets from hydrophobicity to hydrophilicity enables new avenues for their aqueous applications in diverse technological and industrial fields.

The plasma treatment method was extensively researched in a previous study as it is one of the often-used physical methods to assign extrinsic hydrophilicity to CNTs [5,6,7,8]. Plasma treatments have been shown to alter the surface compositions of F and O atoms drastically, which in turn affect the hydrophobic and hydrophilic properties of the CNTs. Unfortunately, these treatments often involve the use of atmospheric-pressure plasma jets, high-voltage transformers, and specific gas-flow controllers, which necessitate more complex, expensive, and bulky experimental setups. Such complexity increases not only the operational challenges but also the overall cost of the treatment process, thus limiting wider application in industrial fields [5,9].

In the present work, the electrochemical wetting method is introduced as a highly effective and simple method for hydrophilic functionalization of the surfaces of CNT sheets. This method employs a straightforward three-electrode electrochemical cell setup, where the CNT sheet acts as the working electrode. A potentiostatically applied voltage to the working electrode immersed in an aqueous solution enables the facile introduction of oxygen-containing functional groups. The simplicity of this setup allows a more user-friendly as well as cost- and time-effective approach, which is particularly advantageous for both research and industrial applications. The proposed electrochemical wetting process can be completed in a relatively short time, and the duration can be easily controlled to tune the degree of surface wetting. Furthermore, electrochemical methods have been demonstrated for effective transition from hydrophobic to hydrophilic states, thereby enhancing the CNT wettability [10,11,12,13,14].

## 2. Materials and Methods

### 2.1. Chemicals and Materials

The CNT (carbon nanotube) sheets were drawn from well−aligned multiwalled carbon nanotube (MWNT) forests grown via chemical vapor deposition (CVD), with a height of 750 μm (NTAD 10, PDSI Corporation, Suwon, Republic of Korea). A Polyethylene terephthalate (PET) film was cut to an area of 2 (width) × 3 (length) cm^2^.

In addition, 0.1 M sodium sulfate (Na_2_SO_4_) aqueous electrolyte was prepared by dissolving 1.42 g Na_2_SO_4_ in 100 mL deionized water. The solution was stirred until it was transparent.

### 2.2. Preparation of the Hydrophobic CNT Sheets/PET (CP) Film

Five layers of CNT sheets drawn from a 300-um-high MWNT forest (A-Tech System Co., Incheon, Republic of Korea) were uniformly and straightly into a 30-mm-long and 10-mm-wide sheet stack on the PET film. The ethanol densification process was performed, resulting in physical adhesion between the CNT sheet and the PET film. For a stable electrochemical wetting (ECW) treatment and electrochemical performance characterization, both the ends of the CNT/PET film (area: 2 cm^2^) were electrically connected to Cu wires (diameter: 180 um) using silver paste and then chemically tethered using an epoxy glue (Devcon, Danvers, MA, USA). The ECW treatment was performed using a three-electrode electrochemical system consisting of Ag/AgCl (reference electrode), CNT sheets/PET film (working electrode), a Pt mesh (counter electrode), and 0.1 M Na_2_SO_4_ (the liquid electrolyte). Then, a potentiostatic voltage (vs. Ag/AgCl) was applied using an electrochemical analyzer (Vertex EIS, Ivium, Eindhoven, The Netherlands).

### 2.3. Characterization

Scanning electron microscope (SEM) images of the CNT and ECW treated CNT sheets/PET film were obtained (S−4600, Hitachi, Tokyo, Japan), along with optical images of the film using an optical camera (D750, Nikon, Tokyo, Japan). Contact and sliding angle value between CNT/PET films and water droplets were measured (Phoenix-MT(M), S.E.O., Suwon, Republic of Korea). In addition, the electrochemical performances of the CNT/PET films were evaluated using an electrochemical analyzer (Vertex EIS, Ivium, Eindhoven, The Netherlands). The chemical composition and features of the CNT sheets were determined via XPS measurements performed (Versaprobe II, ULVAC–PHI, Kazaki, Kanagawa, Japan).

## 3. Results and Discussion

### 3.1. ECW-Treated Carbon Nanotube (CNT) Sheet on a Polyethylene Terephthalate (PET) Film

Pulling a CNT sheet from a CNT forest is a fascinating and intricate procedure that has been optimized for scalability and high production rate [1]. To initiate the drawing process, the sidewalls of the CNT forest are placed in contact with metal sticks, and anisotropically aligned and self-supporting CNT sheets are subsequently hand drawn, as shown in Figure 1a. The sheets are later transferred onto a PET substrate, and five stacks of the CNT sheets are used as the working electrode for the electrochemical wetting treatment in our experiment (Figure 1b). Before treatment, the CNT sheets were cleaned and cut to the desired dimensions. Copper wires were attached to both ends of the sheets to provide electrical connections to apply the constant voltages during electrochemical treatment (Figure 1c). Before applying electrochemical wetting, a pretreatment is required to stabilize the CNT sheets, which is alcohol-based densification. As the CNT sheets are not chemically bonded to the PET substrate, they may detach or aggregate after electrochemical wetting. To prevent this, we dropped alcohol on the sheets and dried them for an hour. This pretreatment helps the sheets to attach well to the PET substrate as the sheets can be highly densified (up to 360 folds) and therefore stabilized on the substrate [1]. Electrochemical wetting of the CNT sheets was carried out in a conventional three-electrode electrochemical cell, as schematically illustrated in Figure 1d. The CNT sheets were the working electrode, while a platinum mesh and an Ag/AgCl electrode functioned as the counter and reference electrodes, respectively. An electrochemical analyzer was employed to control the applied voltage and current throughout the experiment. The working, counter, and reference electrodes were then immersed carefully in an electrolyte of 0.1 M Na_2_SO_4_ aqueous solution (Figure 1e). Potentiostatic voltages of 2, 3, and 4 V were applied to the working electrode, and the corresponding current densities versus wetting times were plotted (Figure 1f). The current density remained relatively stable during the early stages of the wetting process, showing plateau shapes for about 6 and 3 s at 3 and 4 V, respectively. Subsequently, a drastic decrease in current was observed, which may be attributed to the increase in sheet resistance due to excess oxygen functionalization (Appendix A) [10].

To assess the structural integrity after electrochemical wetting, we observed the surface conditions of the CNT sheets before and after treatment (Figure 1g). From the scanning electron microscopy (SEM) images, no observable change was detected. Even at a higher magnification, highly aligned CNT bundles and their network structure were observed without significant structural collapse (Figure 1h). We expect that appropriate surface functionalization of the sheets was achieved by introducing oxygen-containing groups (e.g., hydroxyl groups), which will be discussed later (Figure 1i). The changes in the surface conditions can be simply confirmed by the large decrease in the contact angle of a water droplet placed on the surface after electrochemical wetting: the 112° contact angle of the pristine CNT decreased dramatically to 48° after wetting, as shown in Figure 1j. The change in contact angle in the image satisfies the Young’s equation ($\gamma_{SV}=\gamma_{SL}+\gamma_{LV}COS\left(\theta \right)$), as mentioned by the reviewer [15].

### 3.2. Changes in Contact Angle on the IHCP Surface

To inspect the electrochemical wetting effects, the contact angles between water droplets and the CNT sheets are systematically compared in Figure 2. Before electrochemical wetting, the CNT sheet exhibits a typical hydrophobic behavior characterized by a large contact angle often exceeding 110°. This intrinsic hydrophobicity of the CNT sheet is attributed to the inherent chemical structure of the CNTs, which entails symmetric bonding without any polar functional groups. Upon electrochemical wetting, the contact angle decreases progressively to about 40°, showing effective transition to extrinsic hydrophilicity.

In the electrochemical wetting process, two key parameters, namely applied voltage and wetting time, play important roles in determining the contact angles of water droplets placed on the CNT sheets. In general, the applied voltage and wetting time are inversely related; as the applied voltage increases, the wetting time required for electrochemical CNT activation decreases. The reason behind this inverse relationship lies in the electrochemical kinetics of the process; a higher applied voltage accelerates the electrochemical reactions that introduce oxygen-containing functional groups to the CNT surfaces. Although wetting at a higher voltage induces faster transition to hydrophilicity for the CNT sheet (e.g., 49° contact angle at 4 V over 3 s), wetting at a lower voltage with a longer treatment time can produce similar effects (e.g., 48° contact angle at 2 V over 30 s). However, it should be noted that excess activation can lead to destruction of the π–π conjugate structure of the CNTs, thereby increasing the sheet resistance drastically. Therefore, the applied voltage and treatment time must be carefully optimized to achieve the desired hydrophilicity without degrading the structural integrity of the CNTs. By multiplying the applied voltage and treatment time, the optimized energy can be calculated to be approximately 0.21 J/cm^2^ to fully oxidize the CNT sheets in this work.

### 3.3. Sliding Angle Variation for IHCP and EHCP Film

In Figure 3, based on these findings, one can also expect a substantial change in the sliding angle of the CNT sheets after electrochemical wetting. The initial hydrophobicity of the CNT sheets may likely result in a low sliding angle (42°), requiring a small tilt to initiate sliding of a water droplet. However, after electrochemical wetting, the sliding angle increases dramatically (180°) owing to the enhanced hydrophilicity of the surface, and the pinned state of the water droplet can be observed for all inspected voltages.

As mentioned before, facile transition of the CNT sheet from hydrophobicity to hydrophilicity through electrochemical wetting is a multifaceted process that involves both chemical and structural modifications. From the chemical perspective, electrochemical wetting introduces various oxygen-containing functional groups, such as carboxyl, hydroxyl, and epoxy groups, on the surfaces of the CNTs. These polar functional groups increase the surface energy and wettability of the CNT sheet, thereby improving its hydrophilicity. From the structural perspective, electrochemical wetting involves the infiltration of electrolyte ions between adjacent CNTs within the sheets via double-layer charge injection [16,17]. Therefore, the combination of electrolyte-infiltration-induced internanotube swelling and high water interactivity of the electrochemically wetted CNT sheets contributes to a significant decrease in the contact angle [12].

### 3.4. Characterization as Applied Voltage and Time Correlation among Contact, Sliding Angle and ECW Treatment

The effects of electrochemical wetting on the changes to both contact and sliding angles are plotted in Figure 4a. The prepared pristine CNT sheets, which were densified and delaminated on the PET substrates, had average initial contact and sliding angles of 112° and 42°, which changed to 48° and 180° (pinned state), respectively, after electrochemical wetting, showing effective transition to hydrophilicity. The changes in the contact and sliding angles are plotted versus treatment time in Figure 4b. The higher applied voltage of 4 V achieves the lowest contact angle (44°) over the shortest time (3 s), whereas the lower voltage of 2 V needs more treatment time to produce a similar level of hydrophilicity. The contact angles versus calculated total energies are plotted for the applied voltages of 2, 3, and 4 V and compared (Figure 4c–e). Once a certain voltage level is applied (e.g., above 2 V) and given enough wetting time (or energy), the surface oxidation effects seem to converge regardless of the applied voltage. One of the optimum input energies for assigning extrinsic surface hydrophilicity to the CNT sheet is about 0.21 J for the given voltage range (Figure 4f). This energy value corresponds to 2 V application (30.7 mA maximum current) over 30 s, 3 V application (79.5 mA maximum current) over 6 s, and 4 V application (115.6 mA maximum current) over 3 s. The corresponding contact angles are shown in Appendix A.

### 3.5. Chemical and Electrochemical Characterization of the EHCP Surface after ECW Treatment Process

The electrochemical wetting method has been shown to introduce oxygen-containing functional groups on the surfaces of CNT sheets. These functional groups play pivotal roles in altering the surface properties of the materials and can be characterized using X-ray photoelectron spectroscopy (XPS). This technique provides insights into the types and quantities of functional groups introduced, thereby allowing a comprehensive understanding of the surface chemistry after electrochemical wetting. This change in surface chemistry is a primary factor contributing to the drastic reduction in the contact angle. The newly formed functional groups and altered microstructure collectively contribute to the significant decrease in contact angle, making the surface highly hydrophilic. With increasing oxidation time, the oxygen and carbon (O/C) atomic ratios rose from 0.06 to 0.25, indicating effective control over the oxygenic groups on the surfaces of carbon nanotubes (CNTs) [18]. To investigate the detailed chemical composition of CNT sheets, the XPS survey and the high-resolution C 1s profiles were investigated at different voltages and electrochemical wetting times. When the applied voltage was increased from 2.0 to 4.0 V (vs. Ag/AgCl), the wetting time decreased from 30 to 3 s, indicating that these two parameters are inversely related (each of the left side of Figure 5a–c, respectively). The gray dash line presents the original total profiles of the high resolution C 1s XPS analysis spectra. Also, other colored lines present the detailed surface functional groups content and composition were analyzed by de-convoluting the C 1s peaks (284 eV) using Gaussian peak fitting(each of the right side of Figure 5a–c, respectively). This analysis considered specific bonds: C=C (284.3 eV), C–C (284.8 eV), C–OH (286.2 eV), C–O–C (287.3 eV), and COOH (288.7 eV) [6,7,18,19]. Notably, irrespective of the applied voltage in given range, several bonds were clearly observed at 284 eV (C=C sp2), 284 eV (C–C sp3), 286 eV (C–OH), 287 eV (C–O–C), and 288 eV (COOH). The types of introduced functional groups were also observed in the high-resolution O1s profile in Appendix A (533 eV). The formation of oxygen functional groups is clearly observed in the Raman spectroscopic analysis as well (Appendix A). Both spectra exhibited prominent peaks at ~1347 cm^−1^ (D band) and ~1584 cm^−1^ (G band). Particularly noteworthy is the substantial increase in the intensity of the D band after ECW treatment. Furthermore, the intensity ratio of the D and G bands (*I_D_/I_G_*), which indicates the extent of defect density in the graphitic carbon structure, significantly increased from 0.49 to 0.79. These results imply the presence of defects in the intrinsic carbon bonds within the CNTs as a consequence of ECW treatment.

In Figure 5d, the relationship between the water contact angles and the wetting degree of the CNT sheet is illustrated. Initially, pristine CNT sheets, characterized by an O/C ratio of 0.06, exhibited their inherent hydrophobic nature, displaying a contact angle (CA) of approximately 120°. Through the ECW treatment process, there was a proportional reduction in the CA observed between the water droplets and the sheet surface, reaching a decrease of up to 48° as the O/C ratio increased to approximately 0.25. This result signifies that the CNT sheets treated with the ECW process became significantly more hydrophilic, indicating an enhancement in wettability.

The oxygen-containing groups introduced by electrochemical wetting contribute to a pseudocapacitive behavior in an aqueous electrolyte system, significantly enhancing the capacitance of the resulting sheet-based supercapacitor [11,12,20]. Therefore, in the present case, the electrochemically wetted CNT sheet has a profound impact on capacitance improvement. Before electrochemical wetting, the energy storage mechanism of the CNT-based supercapacitor is primarily based on the electrochemical double-layer capacitance (EDLC) [20,21,22]. However, the specific capacitance based on the EDLC depends only on the surface ion adsorption; therefore, the values may be unsatisfactory for practical applications, especially when compared with supercapacitors containing pseudocapacitive guest materials like conducting polymers or transition metal oxides [23,24,25,26,27,28,29,30,31]. Upon electrochemical wetting, the CNT yarns undergo significant functionalization with oxygen-containing groups, and these groups can be electrochemically active sites for pseudocapacitive charge storage, as reported previously [10,12,13,18,20,32]. Pseudocapacitance arises from the Faradaic redox reactions that occur at the surfaces of the material, in contrast to the non-Faradaic electrostatic storage of charge in the double-layer capacitors.

Therefore, functionalization drastically enhances the performance of the sheet-based supercapacitor. For instance, the capacitance of the electrochemically wetted CNT was found to be 19 times larger than that of the neat CNT sheet, which can be confirmed from the cyclic voltammetry and galvanostatic charge/discharge curves shown in Figure 5e,f, respectively. Moreover, the oxygen-containing groups improve the wettability of the CNT sheets, which is beneficial for electrolyte infiltration. Better electrolyte infiltration ensures that a larger surface area of the CNT is accessible for charge storage, thereby enhancing the capacitance further. Therefore, electrochemical oxidation is a versatile tool for tuning the properties of CNT sheets for energy storage systems [33,34].

The unique network structure of the CNT sheets allows retention of electrical conductivity when the sheets are bent or deformed. The interconnected CNT bundles create multiple pathways for electron flow, ensuring that the electrical conductivity is maintained even when the sheets are bent. The electrochemically oxidized CNT sheets also demonstrate flexibility, where the resistance barely changes after 150 cycles of bending deformations (bending angle = 120°), as shown in Figure 5g. This is particularly important for applications requiring flexibility, such as wearable electronics, flexible batteries, or electrochemical catalysis reactions [35,36].

## 4. Conclusions

This study entails an in-depth exploration of the electrochemical wetting method as an effective strategy for functionalization of CNT sheets. The electrochemical wetting method is known to offer a range of advantages over other functionalization techniques, such as plasma treatment and acid wetting. While plasma treatment necessitates specialized equipment and can be both energy-intensive and time-consuming, acid wetting often involves the use of hazardous chemicals and stringent safety protocols. Electrochemical wetting, on the other hand, is characterized by its simplicity, cost-effectiveness, and efficiency; it employs a straightforward electrochemical setup and does not require specialized gases or chemicals. It is shown that this facile method induces significant transition from hydrophobic to hydrophilic behaviors, as evidenced by the dramatic decrease in the contact angles of water droplets on the CNT surfaces. The introduction of oxygen-containing functional groups and electrolyte-infiltration-induced internanotube swelling are identified as the key mechanisms behind this transition. Moreover, this study demonstrates a remarkable improvement in the supercapacitor performance of the electrochemically wetted CNT sheet.

The environmental sustainability of the electrochemical wetting method is an important consideration, especially in the current global context of growing environmental concerns. While the electrochemical wetting process is relatively straightforward and does not require hazardous chemicals, a thorough environmental impact assessment is still essential. This could involve a lifecycle analysis, evaluating the energy consumption of the electrochemical process, recyclability or disposability of the electrolyte, and any waste byproducts generated. Such assessments would offer a holistic perspective on the environmental sustainability of the electrochemical wetting method, providing valuable data for policymakers, researchers, and industry stakeholders concerned with minimizing the environmental impact.

## Figures and Tables

**Figure 1 nanomaterials-13-02834-f001:**
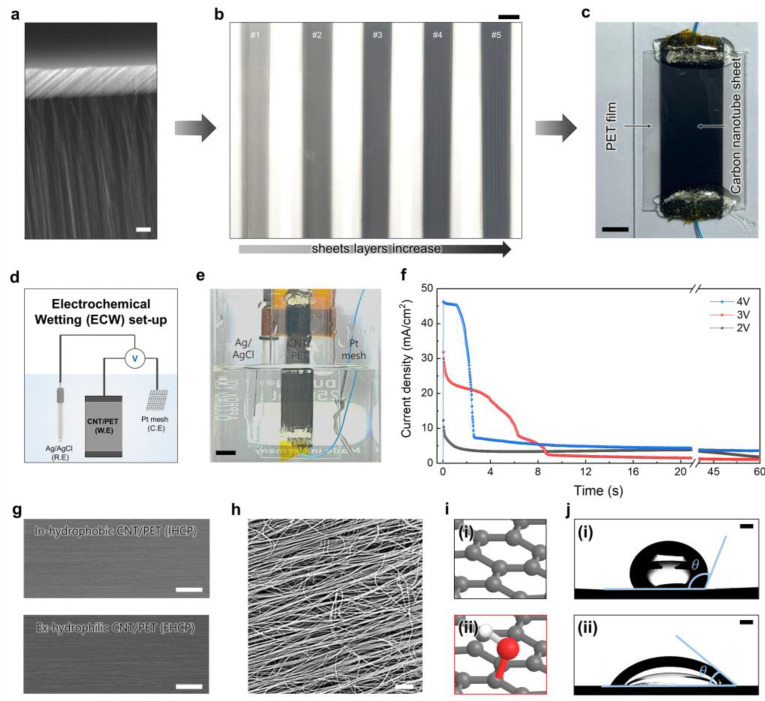
(**a**) Fabrication of CNT sheets/PET film. Scanning electron microscopy (SEM) image of the CNT sheet drawn from the CNT forest (scale bar = 25 μm). (**b**) Photograph showing stacking of the CNT sheets for up to five layers (scale bar = 0.5 cm). (**c**) Prepared CNT sheets loaded on PET film and electrically connected to Cu wires for electrochemical wetting (scale bar = 0.5 cm). (**d**) Schematic illustration and (**e**) photograph showing the setup for the electrochemical wetting treatment based on the three-electrode system consisting of a 20-mm-long CNT/PET film (working electrode), Ag/AgCl (reference electrode), and Pt mesh (counter electrode) in 0.1 M Na_2_SO_4_ electrolyte (scale bar = 1 cm). (**f**) Current density (normalized to the surface area of the CNT sheets) versus wetting time under various potentiostatically applied voltages ranging from 2.0 to 4.0 V (vs. Ag/AgCl). (**g**) SEM images of the surface before (upper panel) and after (lower panel) electrochemical wetting (scale bar = 50 um). (**h**) High-magnification SEM image showing the surface structure of the electrochemically wetted CNT sheet (scale bar = 10 μm). (**i**) Schematic images showing the graphitic atomic structures of the intrinsically hydrophobic CNT sheet (upper panel (i)) and hydroxyl group introduced by electrochemical wetting for extrinsically hydrophilic CNT sheet (lower panel (ii)). (**j**) Contact angle difference between water droplets placed on the pristine CNT sheet (upper panel (i)) and electrochemically wetted CNT sheet (lower panel (ii)); the scale bars for both images are 0.1 cm.

**Figure 2 nanomaterials-13-02834-f002:**
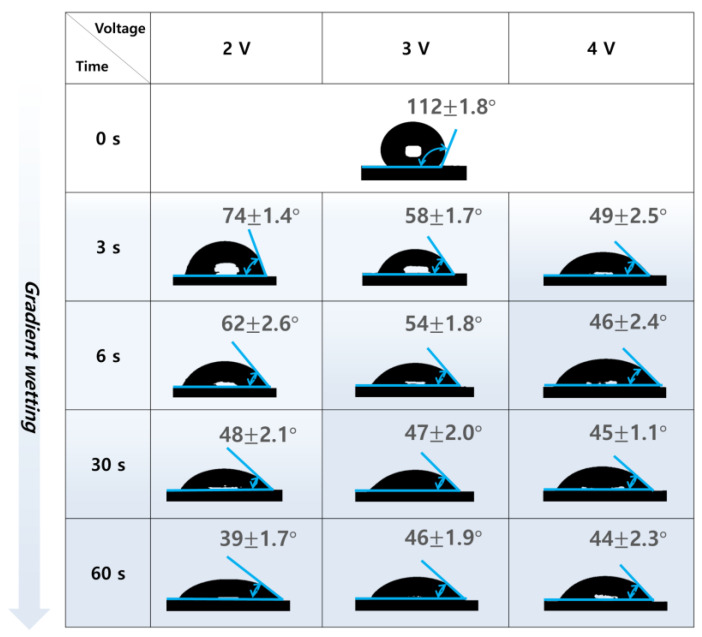
Photographs showing the contact angles of water droplets placed on CNT sheets that were electrochemically wetted at various voltages for various treatment times.

**Figure 3 nanomaterials-13-02834-f003:**
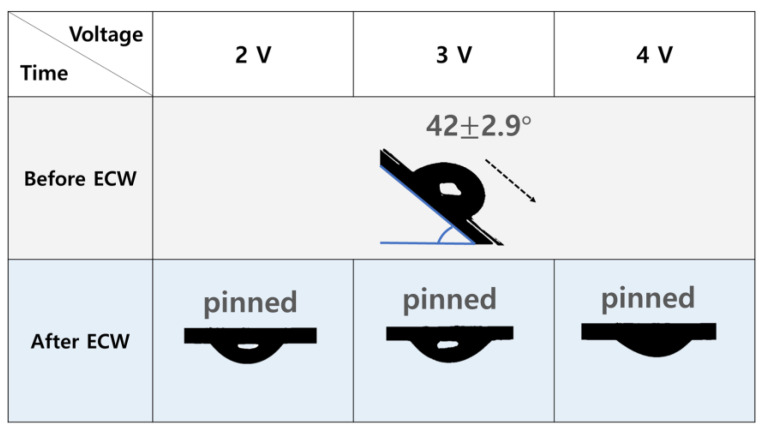
Photographs showing the sliding angles of water droplets placed on the CNT sheets before and after electrochemical wetting under various voltages.

**Figure 4 nanomaterials-13-02834-f004:**
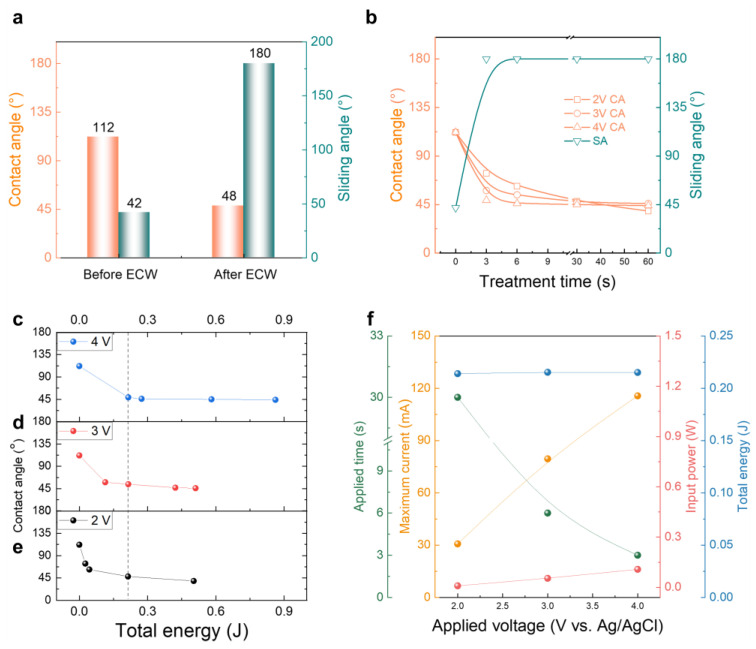
(**a**) Measured contact angles and sliding angle changes before and after electrochemical wetting. (**b**) Contact and sliding angles versus treatment times for various applied voltages. Measured contact angles versus total energy input for potentiostatically applied voltages of (**c**) 4 V, (**d**) 3 V, and (**e**) 2 V. (**f**) Treatment time, maximum current, input power, and total energy are compared versus the applied voltage.

**Figure 5 nanomaterials-13-02834-f005:**
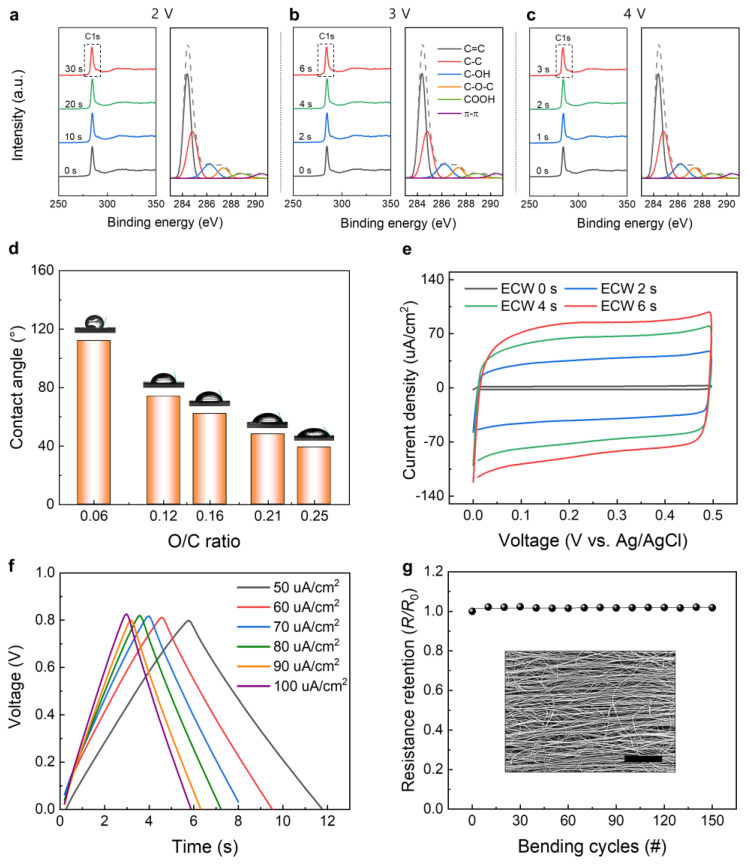
XPS survey spectra (left panel) and high-resolution C 1s (right panel) XPS analysis spectra of CNT sheets for various oxidation times at applied voltages of (**a**) 2 V, (**b**) 3 V, and (**c**) 4 V. (**d**) Oxygen/carbon (O/C) ratios for different applied voltages (2.0–4.0 V) and treatment times (up to 30 s). (**e**) Cyclic voltammetry curves (at 10 mV/s) with a three-electrode system and compared with ECW-treated CP films over applied time (0–6 s). (**f**) Galvanostatic charge/discharge curves of the electrochemically wetted CNT sheets over 50–100 µA/cm^2^ of current density. (**g**) Resistance changes versus bending cycle (bending degree = 120°) for the electrochemically wetted CNT sheets; the inset shows the SEM image of the CNT sheet (scale bar = 10 μm).

## Data Availability

Not applicable.

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
