# Peer review of "Transition of Carbon Nanotube Sheets from Hydrophobicity to Hydrophilicity by Facile Electrochemical Wetting"

_nanomaterials, 2023, doi:10.3390/nano13212834_

Round 1

Reviewer 1 Report

Comments and Suggestions for Authors

In this work, the authors report a cost-effective and time-efficient electrochemical oxidation method to convert CNT sheets from hydrophobicity to hydrophilicity. Also, the mechanistic insights into the hydrophilic transition are provided. Overall, this manuscript is interesting can be accepted after well addressing the following important issues.

1.      For the analysis of contact angle, Young equation should be described and discussed, which is very important for analyzing contact angle. The authors should refer to this work (DOI: 10.1063/5.0083059).

2.      Necessary references should be added to support the reliability and rationality of XPS analysis.

3.      Raman should be further conducted to identify the detailed species in the CNT and treated CNT.

4.      Besides the possible applications in wearable electronics and flexible batteries, this method to prepare hydrophilic CNT can also be used for electrochemical catalysis reactions. The authors should cite some references to support this part, for examples DOI: 10.1039/D3EE02695G and 10.1016/j.apcatb.2021.120484.   

Comments on the Quality of English Language

Minor editing of English language required

Author Response

Response to Reviewers’ Comments

Dear Editor:

We appreciate the comments of the reviewers and the suggestions of the editor. Our responses to these comments are listed below, and the manuscript and supplemental materials have been accordingly revised to clearly address the issues raised by reviewers. To help address the comments of the reviewers, we performed additional experiments and added pertinent new results. The main revision contents include (1) Raman analysis to characterize the functional groups introduced during the electrochemical wetting process, (2) rewriting the recommended parts in the revised manuscript to more clearly explain our results. We performed our best to address the issues raised by reviewers in the limited revision due time. According to the comments from the reviewer, the manuscript was revised using more abundant expressions and sentence patterns.

Thank you for your efforts in evaluating the suitability of this manuscript for publication in Nanomaterials.

Sincerely,                                                                             

Prof. Changsoon Choi

 Dongguk University

[email protected]

Reviewer 2 Report

Comments and Suggestions for Authors

This paper presents the electrochemical oxidation of carbon nanotubes to modify the surface properties of the carbon nanotubes transitioning from pristine hydrophobic to hydrophilic behaviour. Authors employed a facile electrochemical approach to achieve the targeted results which is evidenced primarily by contact angle analysis. This work presents some useful insights for surface modifications of CNTs. However, some concerns should be addressed for better understandings of the approach and applications of CNTs.

1.     Authors used high voltage 2-4V, which converts the surface properties of the CNTs, however, it is vital to analyze the degree of oxidation of the CNTs. Voltage of more than 2 V is generally corrosive to carbon materials and results in extensive oxidation and degradation of the CNTs, therefore, CNTs might inevitably undergo surface oxidation that in general has negative effects on the electrochemical performance. What factors or parameters define the degree of oxidation and how we can achieve the desired targets for certain applications, this needs to be explained.

2.     The XPS survey in figure 5a shows the oxidation behaviour of CNTs at different temperatures, however, it is hard to guess the actual change in C oxidation behavior from rough survey scan. Authors presented a single C in figure 5b, which is not sufficient to analyze the degree of oxidation at different time durations. Authors are suggested to provide XPS results of C1s at different voltages, which is vital for degree of oxidation of CNTs by comparing the surface functional groups content and composition for instance, this work provide quantitative analysis of the surface functional groups after CNTs oxidation. (J. Mater. Chem. A, 2015,3, 7575-7582)

3.     Figure 3 shows the wetting properties at different voltages, where the results after wetting at 2, 3,and 4 V are quite similar. How the voltage affects the wetting properties of the CNTs, authors should enlightens the role of voltage in wetting behavior of the CNTs.

4.     Figure contradicts the statement that ‘’The higher applied voltage of 4 V achieves the lowest contact angle (44º) over the shortest time (3 s), whereas lower voltage of 2 V needs more treatment time.’’ It can be seen from Figure 4b, that after 0 seconds, the A is minimum for 2V, while it is same at approx. 42 seconds.

5.     Authors have made a significant self-citation (half self cited papers), which is not considered a fair literature survey. Authors are suggested to update  the citations with more diversity and recent literature should be cited. 

Comments on the Quality of English Language

NA

Author Response

(The authors gave the same response as above.)

Round 2

Reviewer 2 Report

Comments and Suggestions for Authors

Reviewer is unable to see changes in the revised manuscript as changes are not highlighted in the manuscript.

Authors are suggested to provide a revised manuscript with the changes highlighted.

Please consider inserting XPS figure into the main manuscript and remove some extra contant angle studies into supporting information.

Comments on the Quality of English Language

NA

Author Response

Response to Reviewers’ Comments

Dear editor

We appreciate the comments of the reviewer #2 and the suggestions of the editor. The submitted manuscript and supplemental materials have been accordingly revised to clearly address the issues raised by the reviewer. All the reviewer’ comments and point-by-point responses to reviewer’ questions have been included in the revised manuscript. Specifically, we inserted XPS figure into the main manuscript and remove some extra contact angle studies and relocate into the supporting information. In addition, as requested by the editor, we checked that all references are relevant to the contents of the manuscript. All changes have been highlighted in the “Revised Manuscript (Marked version)” to help the reviewer and editor check the revision.

Thank you for your efforts in evaluating the suitability of this manuscript for publication in Nanomaterials.

Sincerely,

Prof. Changsoon Choi

Dongguk University

[email protected]

Response to Reviewer 2 Comments

Comments and Suggestions for Authors

Reviewer is unable to see changes in the revised manuscript as changes are not highlighted in the manuscript. Authors are suggested to provide a revised manuscript with the changes highlighted.

Answer) We deeply appreciate your continued interest and diligent efforts of the reviewer in assessing our manuscript. We sincerely apologize for any confusion caused by the oversight in our previous revision submission. To address this, we provide a newly revised manuscript with all the changes highlights.

1. Please consider inserting XPS figure into the main manuscript and remove some extra contant angle studies into supporting information.

Response) We appreciate your important comments. As you suggested, the XPS data measured at different voltages and electrochemical wetting times in the supporting information has been inserted into the main figure (Fig. 5a-c). Additionally, extra CA and SA images have been removed from the main figure (Figure 2, Figure 3, and Fig. 4g, h) and relocate to the supporting information (Fig. S2a, b). We have newly added these figures in the revised manuscript.
